# Invasive Aspergillosis after Renal Transplantation

**DOI:** 10.3390/jof9020255

**Published:** 2023-02-15

**Authors:** Liyanage Shamithra Madhumali Sigera, David W. Denning

**Affiliations:** Division of Evolution, Genomics and Infection, School of Biological Sciences, Faculty of Biology, Medicine & Health, University of Manchester, Manchester M13 9PL, UK

**Keywords:** invasive aspergillosis, renal transplantation, immunosuppression, antifungals

## Abstract

Over 95,000 renal transplantation procedures were completed in 2021. Invasive aspergillosis (IA) affects about 1 in 250 to 1 in 43 renal transplant recipients. About 50% of cases occur in the first 6 months after transplantation; the median time of onset is nearly 3 years. Major risk factors for IA include old age, diabetes mellitus (especially if prior diabetic nephropathy), delayed graft function, acute graft rejection, chronic obstructive pulmonary disease, cytomegalovirus disease, and neutropenia. Hospital construction, demolition activities, and residential refurbishments also increase the risk. Parenchymal pulmonary infection is the most common (~75%), and bronchial, sinus, cerebral, and disseminated disease are less common. Typical pulmonary features of fever, dyspnea, cough, and hemoptysis are seen in most patients, but 20% have non-specific general features of illness. Non-specific infiltrates and pulmonary nodules are the commonest radiological features, with bilateral disease carrying a worse prognosis. Bronchoscopy for direct microscopy, fungal culture, and *Aspergillus* antigen are the fastest means of establishing the diagnosis; a positive serum *Aspergillus* antigen presages a worse outcome. Standard therapy includes voriconazole, isavuconazole, or posaconazole, with great attention necessary to assess likely drug–drug interactions. Liposomal amphotericin B and echinocandins are less effective. A reduction in or stopping immunosuppression needs careful consideration, given the overall mortality of IA in renal-transplanted patients; continuing corticosteroid after the diagnosis of IA increases mortality by 2.5 times. Surgical resection or the addition of a gamma interferon should also be considered.

## 1. Introduction

Renal transplantation (RT) is the treatment of choice for patients with end-stage renal failure as it improves the quality of life and survival time of the patient [1]. The world observes a rising number of renal transplantation procedures owing to modern advances in transplant medicine [2]. It is considered the most common solid-organ transplantation type; approximately two-thirds of all solid-organ transplantations (SOTs) [2,3]. For instance, compared to 4300 lung, 5900 heart, and 23,000 liver transplantations, 77,000 renal transplantations were completed in 2012 [4]. In 2021, over 92,500 renal transplantation procedures were recorded by the Global Observatory on Donation and Transplantation, not including China [5]. A successful renal transplant relies on better matching and control of rejection. However, immunosuppressive therapy used paves the way for infections due to viruses, bacteria, and fungi [6]. Fungi are responsible for around five percent of all infections in renal-transplanted patients and *Aspergillus* species is second only to *Candida* spp. in this cohort [7]. Invasive aspergillosis (IA) is a life-threatening opportunistic infection in immunocompromised patients including renal transplant recipients [8]. It renders severe outcomes in the absence of a timely diagnosis and appropriate antifungal management [8].

### 1.1. Incidence

Historically, a low incidence of IA was observed among renal transplant recipients in comparison with other solid-organ-transplanted patients [2,9]. For instance, a multicenter study in France found an incidence rate of IA of 0.4%, 1.3%, and 1.9% after renal, heart, and liver transplantation, respectively [10].

Although the incidence rates are low at 0.5% in most published examples, they range from 0.2% to 14% in different studies [9,10,11]. It is also observed that the reported incidence of IA after renal transplantation varies from region to region across the globe. For example, the reported incidence was 2.3%, 0.4%, and less than 1% in India, France, and England, respectively [8].

### 1.2. Disease Burden (Mortality and Morbidity)

Despite its low incidence, it can be hypothesized that renal transplants have the highest burden of IA in SOTs since the number of renal transplants greatly surpasses the number of other SOTs [10].

A high mortality rate has been observed in IA in renal transplants (40–60%) [1,12]. In some studies, it is as high as 67% to 92%; however, this may be due to the small number of cases of these studies [7,11,13]. For example, Gavalda et al. observed the death of 7 out of 10 renal transplant recipients with IA in his retrospective case–control study, a 70% mortality [13]. Studies observed increased mortality in line with a delay in the diagnosis of IA and a baseline high fever [7]. In addition, some studies have observed significantly higher mortality rates in those with IA early after transplanation compared to those occurring later [2]. However, the true burden of mortality of IA seems to be underestimated among renal-transplanted patients [8].

Fungal infections, including IA, after renal transplantation are also associated with a substantial degree of morbidity [6]. Prolonged hospitalization is a major adverse outcome related to IA in this cohort, and this was observed by Abbott et al. in 2001 [14]. His retrospective evaluation of 33,420 renal transplant recipients in the United States between 1994 and 1997 revealed that the mean length of the hospital stay for kidney-transplanted patients was prolonged in those with IA compared to all other fungal infections (both mold and yeast) [14]. Moreover prolonged hospital stays, the use of expensive antifungals and intensive care in patient management invariably contribute to increased healthcare costs [15].

The management of refractory IA frequently requires the interruption or discontinuation of immunosuppressive drugs. However, this may lead to graft rejection or delayed graft function requiring a return to dialysis [1,16]. For example, a study conducted among 51 renal-transplanted patients with early IA in 19 institutions (from 2000 to 2013) observed graft loss among 25% of survivors (39% died) [10].

### 1.3. Risk Factors for IA in RT

Risk factors for IA among renal-transplanted patients have been evaluated in different studies. A retrospective study conducted at a tertiary-care referral hospital in Korea evaluated patients with invasive pulmonary aspergillosis (IPA) after renal transplantation from February 1995 to March 2015 and found that old age, diabetes mellitus, delayed graft function, and acute graft rejection were associated with the development of IPA [2]. A multinational case–control study conducted by López-Medrano et al. further added chronic pulmonary obstructive disease and the development of bloodstream infection as risk factors for IPA among RT [4]. In his later study, López-Medrano et al. further concluded that underlying diabetic nephropathy is a risk factor [17]. Altiparmak et al. also concluded that old age, prolonged antibiotic course, cytomegalovirus disease, neutropenia, and anti-rejection therapy, such as pulse steroids and anti-lymphocytic globulins, are risk factors [9]. Apart from that, ongoing construction or demolition activities in hospital premises or residential areas also increase the risk [18].

### 1.4. Timing of Infection

It has been observed that the IA risk is highest within the first 6 months after transplantation. Almost half (43%) of the cases of IPA were diagnosed within the first 6 months after transplantation in a multinational retrospective study conducted in Europe among 112 renal transplant recipients diagnosed with probable (75% of cases) or proven (25%) IPA between 2000 and 2013 [4]. A retrospective study conducted at a tertiary-care referral hospital in Korea among patients with IPA after renal transplantation from February 1995 to March 2015 found that the median time of diagnosis was 161 days [2]. A study that followed up 120 kidney recipients for one year in the Organ Transplant Center in Kuwait from March 2016 to October 2019 observed IA most frequently between 1 and 6 months after transplantation [19]. In addition, the highest incidence of IA was observed in the first 3 months after transplantation in a single-center retrospective analysis of forty cases of IA after kidney transplantation [20].

However, the risk is continuous if patients experience rejection episodes and some studies showed late IPA is as common as early IPA in RT recipients, suggesting that IPA can occur in any period after RT [2]. A retrospective study conducted at a tertiary-care referral hospital in Korea evaluated patients with IPA after renal transplantation from February 1995 to March 2015 [2]. They found that approximately half of IPA in RT recipients developed during the late post-transplant period (>6 months) [2]. Moreover, a single-center, retrospective observational study of 438 patients who underwent renal transplantation between 2010 and 2016 in Turkey observed most IPA infections after the first year of transplantation; the median time to onset of IPA was 32.4 months [7].

### 1.5. Clinical Picture

The lung and the sinuses are the foremost sites of infection, followed by dissemination to other organ systems, mainly to the central nervous system (CNS) [21]. This statement derives from a single center retrospective analysis of forty cases of IA after RT, which observed thirty IPA, four invasive bronchial aspergilloses, one cerebral abscess, and five disseminated cases [20].

IPA may cause necrotizing, rapidly progressive pneumonia along with cavitation, vascular invasion, and hemorrhagic infarcts [6,8]. Patients with IPA may present with fever, dyspnea, cough, and hemoptysis [6]. However, sometimes IPA in renal transplant recipients may present with non-specific symptoms. For example, a multinational retrospective study in Europe observed that one-fifth of the IPA patients presented without typical symptoms of lung infection [4].

The hematogenous spread of *Aspergillus* into the brain resulting in hemorrhagic infarction and abscess has been observed after RT [22]. The clinical picture mimics any other type of space-occupying lesion [22]. The presenting symptoms include stroke-like symptoms or convulsions with or without fever [22]. However, it is worth remembering that other organs system can also be affected, and several other clinical forms of invasive aspergillosis could be seen. For example, rare clinical forms of invasive aspergillosis including *Aspergillus* endophthalmitis, *Aspergillus* pseudoaneurysm in graft sites, *Aspergillus* spondylitis, *Aspergillus* prostatitis, *Aspergillus* thyroiditis, *Aspergillus* peritonitis, hepatic aspergillosis, *Aspergillus* tracheobronchitis, and *Aspergillus* endocarditis after dissemination have been reported.

### 1.6. Diagnosis

The diagnosis of IA is based on a combination of clinical, microbiological, and radiological data [23]. The diagnosis of IA in transplanted populations is difficult and often delayed, leads to a worse prognosis [7,8]. The delayed diagnosis of any invasive mold infection is associated with a high mortality rate [24].

#### 1.6.1. Histology, Direct Microscopic Examination, and Culture

Isolation in culture, direct microscopic examination, and demonstration of *Aspergillus* fungal filaments on histology of respiratory samples and biopsy tissues from involved sites are important steps in diagnosing IA [15,18,25]. Cultures (if positive) allow the isolation of *Aspergillus* species and antifungal sensitivity testing, which is a guide to optimal antifungal management. However, cultures of respiratory origin show only a moderate sensitivity and they have a long turnaround time [26]. According to the study of Brown et al., the specimens from the lungs such as lung biopsy, bronchoscopy, and tracheal aspirates of infected liver and kidney transplant patients are more frequently positive than body fluids such as pleural fluid and ascites fluid [27]. The most commonly isolated *Aspergillus* spp. is *A. fumigatus* from this group of patients [27].

Open lung biopsy, CT-guided transthoracic biopsy, transbronchial biopsy, and video-assisted thoracoscopic surgery provide specimens from sterile sites for culture and histopathology for the diagnosis of IA [23]. In addition, the diagnosis of IPA can be facilitated by bronchoscopy to view the airways in cases of tracheobronchial disease along with bronchoalveolar lavage to detect *Aspergillus* antigen and bronchial aspirate for microscopy and culture [18,28]. However, many of these patients are not suitable candidates for these diagnostic procedures owing to multiple comorbidities. Cerebrospinal fluid examination shows that minimal abnormalities and the CSF culture is rarely positive, throwing a diagnostic challenge for CNS aspergillosis in this group [22], although the *Aspergillus* antigen is usually positive.

#### 1.6.2. Use of Galactomannan ELISA Assay

Galactomannan (GM) is a fungal cell wall carbohydrate that is released during tissue invasion and is widely used in the diagnosis of invasive aspergillosis [29]. However, the unsatisfactory performance of *Aspergillus* galactomannan assay in serum for the diagnosis of IA in solid-organ recipients has been observed [30]. The sensitivity and specificity of serum GM in solid-organ-transplanted individuals are 22% and 84%, respectively [15]. Pfeiffer CD et al. observed a slightly higher pooled sensitivity of 41% of the GM test in solid-organ transplant recipients in their meta-analysis [31]. This finding was further supported by Heylen et al. (2015) who observed a positive GM only in only one-third of invasive aspergillosis patients with a transplanted kidney in a single-center study on invasive aspergillosis after RT [20]. López-Medrano et al. observed 60% of positive GM levels in a multinational cohort study of IPA in RT [4]. Moreover, there are instances of false-positive GM test results reported in renal-transplanted patients, with IA suspected as leading to an incorrect diagnosis [32] and the possibility of reporting bias.

The variable sensitivity of GM observed in different studies creates uncertainty in its use as a diagnostic tool in RT patients. However, certain studies have obtained promising results highlighting the prognostic significance of the GM test. For instance, Heylen (2015) observed that the magnitude of the GM index (optical density > 2) mirrors the mortality of patients [12]. López-Medrano et al. also found that the GM indices were significantly higher in non-survivors at 6 weeks (serum and BAL) compared to survivors [4]. Balcon et al. also observed a link between high mortality and higher BAL galactomannan levels in renal-transplanted patients with IA [7]. This is further supported by a retrospective study conducted at a tertiary-care referral hospital in Korea, which evaluated patients with IPA after renal transplantation from 1995 to 2015 [2]. They observed that a serum GM index of >2 and bronchoalveolar (BAL) GM index of >5.0 was associated with high 12-week mortality rates [2]. Seok et al. concluded that the serum GM level might be used as a predictor of prognosis in renal-transplanted patients with IPA [2]. A study conducted among 1762 solid-organ-transplanted patients observed nine renal-transplanted patients with IA (0.5%), and those with a positive serum GM antigen had a higher mortality [33]. Moreover, a single-center, retrospective study on 438 renal-transplanted patients from 2010 to 2016 in Turkey observed a tendency toward high mortality with a detectable GM in serum [7].

Meanwhile, some authors have suggested testing for GM before transplantation in recipients who will require aggressive immunosuppression [11].

#### 1.6.3. (1,3)-Beta-D-glucan

(1,3)-beta-D-glucan is a cell wall component of fungi and is considered a pan-fungal marker for many fungal pathogens including *Aspergillus* species [30]. The positive predictive value of beta-D-glucan for the diagnosis of an invasive fungal infection among solid-organ-transplanted individuals is moderate [34]. Its usage for the diagnosis of IA in renal transplantation has not been fully investigated. However, a high level of serum (1,3)-beta-D-glucan was observed in one renal-transplanted patient with invasive aspergillosis [35].

#### 1.6.4. Radiology

Imaging evidence of IPA in renal-transplanted patients is often non-specific and may also overlap with other pathologies in the early stages. Radiological appearances vary from non-specific infiltration to nodular opacities, nodular opacity plus infiltrate, halo signs, and cavity formation [7,8]. Macro-nodule(s) (more than 1 cm) along with a halo of ground glass attenuation are considered the classical signs of IPA [23]. However, a multinational retrospective study of 112 renal transplant recipients in Europe between 2000 and 2013 found that well-circumscribed nodules were the most common radiological sign (70%) in IPA, with halo signs seen in 25% [4]. Moreover, bilateral lung involvement was an independent predictor of non-survival in this study [4]. In contrast, a single-center, retrospective observational study of 438 renal transplant recipients between 2010 and 2016 observed infiltration without nodular opacity as the most commonly observed CT image in IPA [7]. However, the same study found more nodular opacities with or without infiltration in IPA compared to non-IPA cases in this cohort [7]. These findings were in line with the findings of a retrospective analysis of cases of invasive fungal infections in renal transplant recipients in a university hospital from 1995 to 2013 [8]. It revealed that renal-transplanted individuals with invasive fungal disease had micro-nodules and alveolar condensation [8]. In addition to the above-mentioned features, pleural effusion is rarely observed in IPA [36].

Since IPA may be followed by CNS aspergillosis, the radiological evaluation (CT or MRI) of IPA patients with neurological symptoms is important. *Aspergillus* brain abscesses may be solitary or multiple ring-enhancing lesions and this was demonstrated by a retrospective study of brain CT and MRI studies of patients with neurological symptoms after liver or kidney transplantations [37]. They observed ring-enhancing lesions, especially in the grey-white matter junction along with central diffusion restrictions on an MRI of the brain [37].

Apart from CNS dissemination, pulmonary aspergillosis may disseminate to several other body sites, so radiological investigations should be conducted accordingly. For example, a rare case of *Aspergillus* spondylodiscitis has presented as a discitis, epidural abscess, and subchondral T2 hypointense band on the lumbar MRI [37].

Figure 1 and Figure 2 shows the Chest X-ray and HRCT images of 58-year-old renal transplant recipient with invasive aspergillosis respectively.

### 1.7. Management

Recommendations for the management of IA are extrapolated from the other groups of immunocompromised groups because of the scarcity of any prospective clinical trials that address the management in RT. The management of IA in renal transplant recipients requires a multidisciplinary approach including selecting the appropriate antifungal therapy and consideration of adjunctive immunotherapy, adjunctive surgical therapy, and a substantial reduction of the immunosuppressive drugs used to prevent rejection.

A key early decision in managing IA in renal transplant recipients is whether to stop or greatly reduce immunosuppression, which will usually result in a loss of the graft [38]. The risk of death is high as the diagnosis is often made late, and so this critical decision is difficult and complex and should be made immediately after IA diagnosis [39]. Corticosteroid use after the diagnosis of IA increases mortality in IA by 2.5 times (in all patient groups) [40]. The use of adjuvant gamma interferon may allow for both patient and graft survival [41], but it is still not always successful and the data supporting this strategy are limited.

#### 1.7.1. Antifungals

Antifungal treatment should be started as soon as possible, as a delay in starting antifungal treatment is associated with a worse prognosis [7]. Voriconazole, lipid-based amphotericin B, isavuconazole, micafungin, and caspofungin are the most frequently used antifungals in the management of IA [15]. Given the risk of drug–drug interactions and renal toxicity, the choice of antifungal therapy in renal transplant recipients requires meticulous guidance. Avoiding nephrotoxic medication is advisable during the management of IA in renal-transplanted patients.

Among the azoles, voriconazole, isavuconazole, and posaconazole are the most frequently used antifungals for IA [42]. Since the *Aspergillus* species is not susceptible to fluconazole, the use of fluconazole is ineffective [30].

##### Voriconazole

Voriconazole is one drug of choice in the management of IA and its superiority above other antifungals including polyenes is well-documented [15,30]. In a global aspergillosis study, voriconazole was found to be ~20% more effective in terms of survival, treatment response, and drug interactions when compared to conventional amphotericin B [43]. In follow up, real-life studies, voriconazole was also more effective than echinocandins [44,45,46]. Liposomal amphotericin B is equivalent to conventional amphotericin B in IA but less nephrotoxic, and so is preferred if amphotericin B is selected [30,47]

However, the administration of voriconazole is not free from adverse effects. Approximately 30% of patients on voriconazole develop a reversible visual disturbance and some patients experience an elevation of liver enzymes [15]. The patient’s liver enzyme concentration should be monitored before commencing therapy and then every 2–4 weeks during therapy [15]. Since prolonged voriconazole therapy is associated with cutaneous squamous cell carcinoma in those with light-colored skin, patients should be advised to avoid sun exposure [15]. In addition, prolonged usage of voriconazole leads to the accumulation of fluoride and is linked to painful periostitis in solid-organ transplant recipients [15]. Apart from that, the accumulation of cyclodextrin of intravenous voriconazole might impair renal function, and its use in significant renal impairment is contraindicated. 

Therapeutic drug monitoring (TDM) of voriconazole is important to guide the dose in the management of IA, owing to its variable degree of first-pass metabolism [48]. The first-pass metabolism of voriconazole is mediated by cytochrome 450 enzymes (CYP2C19, −2C9, −3A4, and −3A5), which show genetic heterogeneity. High serum concentrations are observed in CYP2C19 poor metabolizers [42]. About 3% of Europeans and 15% of Asians are slow metabolizers. Its significant variable pharmacokinetics is also related to hepatic dysfunction, drug–drug interactions, and patient age [42]. Consequently, dose adjustment of voriconazole according to TDM is recommended, especially for patients who do not respond to therapy and for patients with possible toxic effects [15].

The administration of voriconazole in renal-transplanted patients requires an appreciation of the significant interactions between different immunosuppressive drugs (calcineurin inhibitors and mTOR inhibitors) [39]. Since voriconazole is a potent inhibitor of CYP3A4, it has the potential to increase the blood level of calcineurin inhibitors (cyclosporin and tacrolimus), thus leading to toxicity [39,42]. Some drugs, such as sirolimus, are contraindicated with the use of voriconazole [15]. Concomitant use of voriconazole and tacrolimus requires an immediate tacrolimus dose reduction by 66–75% and monitoring the trough level of tacrolimus [42]. In addition, voriconazole interacts with prednisolone and so prednisolone dose reduction is needed.

##### Isavuconazole and Posaconazole

Both isavuconazole and posaconazole are effective and safe azole antifungal agents for IA [29]. Isavuconazole shows a similar efficacy with less liver toxicity and fewer drug interactions compared to voriconazole, as shown in a recent randomized, prospective, double-blind, double-dummy, controlled trial [49]. Although isavuconazole is only a moderate CYP3A4 inhibitor, there are some key drug interactions to be aware of, including tacrolimus and sirolimus.

Posaconazole is also non-inferior to voriconazole in terms of the mortality of IA, as shown in another recent randomized, prospective, double-blind, double-dummy, controlled trial [50]. Moreover, patients had fewer side effects compared to voriconazole, and posaconazole was better tolerated [50]. However, posaconazole therapy also requires the regular checking of electrolyte levels and liver function [50]. 

##### Itraconazole

Itraconazole has been used in the management of IA in certain case reports with variable results. IPA patients treated with itraconazole had a 100% success rate in early case series [38]. However, itraconazole is regarded as a third-line drug therapy due to its variable bioavailability and drug–drug interactions with immunosuppressive therapy. A retrospective evaluation of systemic fungal infections in 296 renal-transplanted patients from 1986 to 1999 observed that all three IPA patients treated with oral itraconazole died [9]. However, itraconazole prophylaxis of 400 mg per day seems to reduce the risk of IA in solid-organ-transplanted recipients, but is not strongly recommended in renal transplantation, because IA is so infrequent [15]. It could be considered in the setting of an outbreak. The cyclosporin dose should be halved on the first day of itraconazole treatment and then measured frequently. 

##### Polyenes

Amphotericin B is recommended as a second-line therapy of IA (amphotericin B lipid complex 5 mg/kg/day IV, liposomal amphotericin B 3 mg/kg/day IV) [15]. A retrospective evaluation of systemic fungal infections in 296 renal-transplanted patients from 1986 to 1999 observed the survival of three out of five patients who were treated with amphotericin B, resulting in a 60% favorable response [9]. Liposomal- or lipid-associated amphotericin B is preferred to minimize nephrotoxicity. Although conventional amphotericin B (0.5–1.0 mg/kg) has been used as a salvage therapy in the absence of lipid-based amphotericin B, a successful therapeutic outcome usually requires the cessation of immunosuppressive therapy and resection of the transplanted kidney.

##### Echinocandins

Echinocandins are partially effective and safe agents for refractory IA [29]. Both micafungin and caspofungin are considered as second-line therapies for IA [15]. They have a minimal toxicity profile and few drug interactions. Echinocandins are frequently used as a combination therapy in refractory IA [15] and they have value in those infected with azole-resistant strains of *Aspergillus*. In addition, dose adjustment during renal insufficiency is not needed for echinocandins.

##### Combination of Antifungals

Some *Aspergillus* species are resistant to antifungals and are refractory to treatment. For example, within the *Aspergillus* species, *Aspergillus calidoustus* and some *A. fumigatus* strains are azole-resistant, while *Aspergillus terreus* and *Aspergillus nidulans* are amphotericin B-resistant [23]. A combination of drugs (e.g., liposomal amphotericin B and echinocandin) may be considered in refractory IA due to the resistant *Aspergillus* spp. [51]. The combination of echinocandin with triazole is also promising; however, this requires further studies [29]. A combination of voriconazole and echinocandin should be reserved for salvage therapy in treatment-refractory cases unless resistance is demonstrated [15]. 

##### Duration of Antifungals

The optimal duration of antifungal therapy for IA in renal-transplant recipients has yet not been established and it should be guided by the extent of the disease, response to therapy, and whether immunosuppression is necessary [15]. Some authors recommend six–twelve weeks of administration of antifungals [15], while others recommend antifungals until two weeks after major clinical and radiological improvement. Some authors recommend 6 months of treatment with voriconazole after surgical treatment of IA [11]. Monitoring of GM and direct smear and cultures could be used to direct the decision of therapy discontinuation [15]. The American Society of Transplantation guidelines recommend antifungals (voriconazole) up to complete clinical improvement followed by secondary lifelong prophylaxis [52]. 

#### 1.7.2. Surgical Treatment

Surgical therapy is an adjunctive approach to treatment with antifungal therapy in the management of IPA in RT patients [52,53]. Early and thorough surgical intervention, along with the use of appropriate antifungals and a substantial reduction of the immunosuppressive drugs apparently contributed to the prolonged survival of RT patients with IA [53]. The surgical resection of necrotic tissue reduces the load of fungi and may allow early discontinuation of antifungal therapy [18,53]. If resection is achieved, it should be complete, if technically possible [52].

Surgical therapy is advocated for localized pulmonary lesions without extra-pulmonary dissemination [52,53]. It is also lifesaving in the treatment of patients with massive hemoptysis [15,29].

The value of adjunctive surgical therapy has been highlighted and linked to the survival of patients in several case reports [11,53,54]. A retrospective study of invasive fungal infections among 296 renal-transplanted individuals emphasizes successful outcomes following surgical intervention, management of super-infection, nutritional and metabolic support, all combined with appropriate antifungal therapy [9]. In some cases, the surgical removal of renal graft has been required due to graft failure [11]. However, it is noteworthy to remember that the patients in this cohort are poor surgical candidates and surgical intervention in this group is often challenging for multiple reasons [52].

#### 1.7.3. Immunomodulation (GCSF/Gamma Interferon)

Recent studies have observed the positive impact of adjunctive immunotherapy along with gamma interferon (IFN-γ) in invasive fungal infections through enhancing host defense mechanisms [55]. IFN-γ has been trialed in renal transplant recipients with life-threatening, refractory invasive fungal infections with positive outcomes [41]. In this case, IFN-γ was not associated with renal allograft dysfunction, IFN-γ toxicity, or IFI relapses during follow-up [41]. A 6-week combination of IFN-γ and antifungals cured infections and led to savings in healthcare costs [41].

### 1.8. Prevention

Most of the preventive methods have been extrapolated from other similar settings because there are no specific recommendations pertaining to the renal-transplanted individuals. Construction in hospital premises or probably at home poses the risk of obtaining invasive aspergillosis [6,56]. Consequently, avoiding such risk areas could be advisable. In addition, by restricting the bringing dust and plant pots into patient rooms would be good practice because they could contain numerous *Aspergillus* spores [6].

The prophylactic use of antifungals in renal transplant recipients remains controversial, attributed to its low incidence, significant drug–drug interactions, inadequate solid evidence, risk of resistance, risk of breakthrough infection, and costs [57]. At the moment, there is no formal recommendation regarding antifungal prophylaxis in renal transplant recipients [4].

## Figures and Tables

**Figure 1 jof-09-00255-f001:**
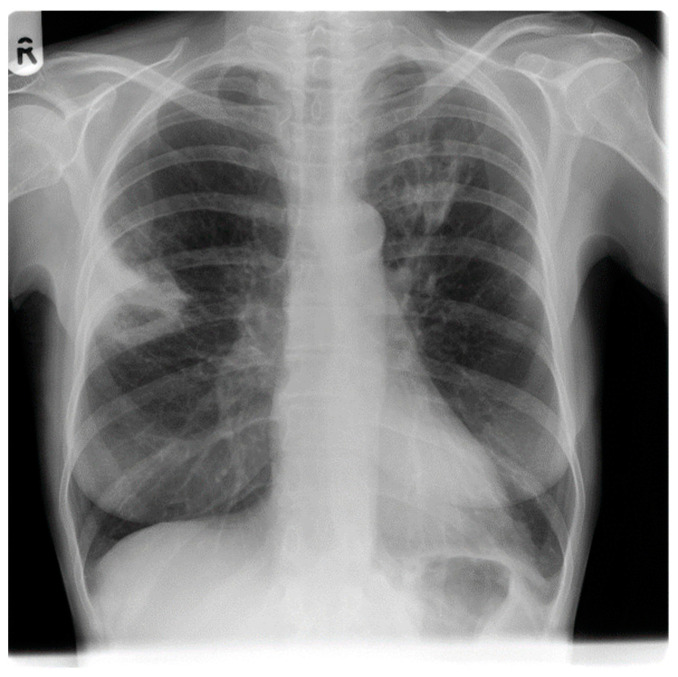
Chest X-ray of invasive aspergillosis in 58-year-old renal transplant recipient shows bilateral infiltrates.

**Figure 2 jof-09-00255-f002:**
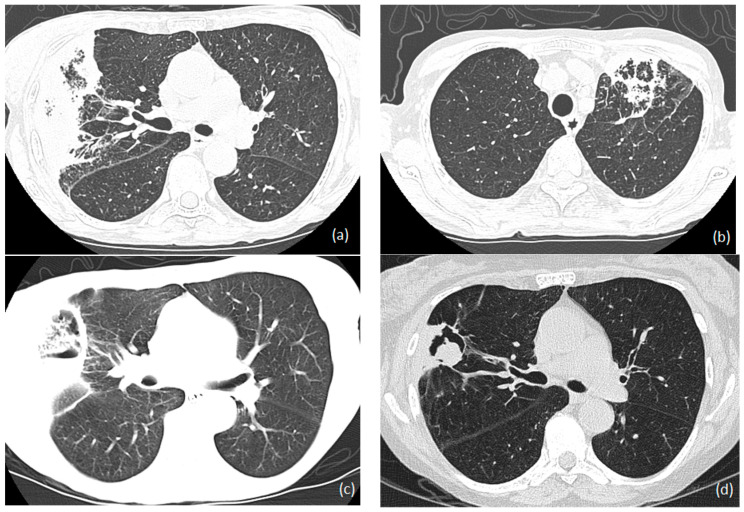
HRCT images of invasive aspergillosis in 58-year-old renal transplant recipient show (**a**) right-sided wedge-shaped consolidation with ground glass changes in the right lower lobe, (**b**) left upper lobe patchy consolidation with ground glass changes laterally, (**c**) thin-walled cavitation with fluid level and material in the cavity in the right lower lobe after 6 weeks of voriconazole, and (**d**) residual findings of the right lower lobe after 2 years of initial presentation and voriconazole treatment showing thin-walled cavity medially, an aspergilloma, and pleural thickening.

## Data Availability

No new data were created or analyzed in this study. Data sharing is not applicable to this article.

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
