# Peer review of "Invasive Aspergillosis after Renal Transplantation"

_jof, 2023, doi:10.3390/jof9020255_

Round 1

Reviewer 1 Report

Generally, well written and covers the topic

Some areas that could be improved

Line 55 sentence” It seems that the incidence of IA after RT depends on the infection control

measures practiced by the different transplantation centres.’   This is a vague statement and the reference just provides soft evidence that  transplantation in India and Russia was associated with higher rates of IA identified in Turkey. If there is evidence this was related to infection control practices this should be referenced. What is this evidence?  

Line 61 – Could some numbers to added to give context to what the high mortality rate is – percentage is a poor indicator in small studies.

Line 65 – the burden referred to her is death I presume – that is only one marker of the burden of disease – clarify

Line 65 Which cohort is the author referring to ?

Line 72 – doses this refer to yeast as well as filamentous fungi – clarify

Line 76 – the reference is a review article – IF there is evidence for resistance developing please provide the primary source

Line 117 – Turkey rather than Turley I think

Line 153 – It unclear what this means – Which lung specimens do you mean ?  BAL samples, expectorated sputum, Induced sputum or  biopsy compared with what - blood cultures and CSF?  

Please clarify

Line 163 – Which aspergillus antigen test do you mean?

Line 194 Turkey again presumably

Line 203 I think reference 35 refers to a single patient – please clarify

Line 215 – Turkey again !!!!

Line 270 presumably you mean - Aspergillus spp are not susceptible to fluconazole so that this is an inappropriate choice for treatment.   It may well act on it in ways that are not obvious.

Line 272 – do you mean that Voriconazole is superior to all other agents or just the polyenes. Your later statement’s on posaconazole and isavuconazole suggest they are superior. Please clarify

Line 280 do you mean concentrations in the peripheral blood rather than levels. Concentration is preferred when the volume of the specimen being analysed is accurately known. Level is useful for the JVP.

Line 294 – What is north Asian ancestry? Can this statement be referenced? Asian is not really an ethnicity.

Line 327 How many patients were treated with Itraconazole – do you mean all treated with itraconazole died?

Line 336 – 50% of how many treated with polyenes had a favourable response?

Line 355 What is the role of combination therapy in infections other than A. terreus and A nidulans or should this statement refer to all Aspergillus species.

Author Response

11/02/2023

Dr.L.S.M.Sigera

Manchester Fungal Infection Group,

Core Technology Facility,

The University of Manchester,

Manchester Academic Health Science Centre,

Manchester, UK.

Dear Reviewer of the Journal of Fungi,

Invasive aspergillosis after renal transplantation (Manuscript ID jof-2189704)

I am submitting a revised manuscript for consideration for publication in the special Issue "Biology, Immunology, Epidemiology, and Therapy of Fungal Infections: A Themed Issue Dedicated to Professor David A. Stevens". The manuscript is entitled “Invasive aspergillosis after renal transplantation”.

We really appreciate your comments because they invariably improve the quality of the manuscript. All the changes were highlighted (In Yellow) and comments were answered in the original manuscript as track changes.

……………………….

Line 55 sentence” It seems that the incidence of IA after RT depends on the infection control measures practiced by the different transplantation centres.’   This is a vague statement and the reference just provides soft evidence that  transplantation in India and Russia was associated with higher rates of IA identified in Turkey. If there is evidence this was related to infection control practices this should be referenced. What is this evidence? - Thank you for the suggestion. We removed the above sentence from the article. 

  • Line 61 – Could some numbers to added to give context to what the high mortality rate is – percentage is a poor indicator in small studies.- Added. Corrected  
  • Line 65 – the burden referred to her is death I presume – that is only one marker of the burden of disease – clarify Mortality .Corrected  
  • Line 65 Which cohort is the author referring to ? Renal transplanted patients. Corrected
  • Line 72 – doses this refer to yeast as well as filamentous fungi – clarify Both yeast and moulds .Corrected
  • Line 76 – the reference is a review article – IF there is evidence for resistance developing please provide the primary source- We would like to remove this sentence. 
  • Line 117 – Turkey rather than Turley I think-Corrected
  • Line 153 – It unclear what this means – Which lung specimens do you mean ?  BAL samples, expectorated sputum, Induced sputum or  biopsy compared with what - blood cultures and CSF?  -Specimens from the lungs (lung biopsy, bronchoscopy and tracheal Body fluids such as pleural fluid and ascites fluid- corrected
  • Line 163 – Which aspergillus antigen test do you mean?-ELISA. Corrected
  • Line 194 Turkey again presumably-Corrected
  • Line 203 I think reference 35 refers to a single patient – please clarify- Single patient. Corrected
  • Line 215 – Turkey again !!!! -Corrected 
  • Line 270 presumably you mean - Aspergillusspp are not susceptible to fluconazole so that this is an inappropriate choice for treatment.   It may well act on it in ways that are not obvious.-Corrected
  • Line 272 – do you mean that Voriconazole is superior to all other agents or just the polyenes. Your later statement’s on posaconazole and isavuconazole suggest they are superior. Please clarify- Voriconazole is superior to polyenes. Corrected
  • Line 280 do you mean concentrations in the peripheral blood rather than levels. Concentration is preferred when the volume of the specimen being analysed is accurately known. Level is useful for the JVP.-Levels changed to concentration .Corrected  
  • Line 294 – What is north Asian ancestry? Can this statement be referenced? Asian is not really an ethnicity.-Asian ancestry was changed to Asians .Corrected  
  • Line 327 How many patients were treated with Itraconazole – do you mean all treated with itraconazole died?-3/3 died. Corrected
  • Line 336 – 50% of how many treated with polyenes had a favourable response?-3/5 survived.Corrected
  • Line 355 What is the role of combination therapy in infections other than  terreusand A nidulans or should this statement refer to all Aspergillus species- The sentenced was rearranged.

Thank You

Yours sincerely

………………………..

Dr.L.S.M.Sigera

Reviewer 2 Report

Its’ well written, nicely complied the published data.

Study presents the main points:

Infectionpulmonary aspergillosis  in renal transplant patient get infections after 6 months

Can present without typical sign symptoms of lung

It can present like space occupying lesion

Infarctions, convulsion, tracheobronchitis endocarditis after dissemination and few others

Like other IPA high GM index values predict high mortality

Radiological finding: Most of the patient present well circumscribed nodules

Pleural involvement is rarely seen

Treatment: other than antifungals gamma interferon allows both the patients and graft survival

After treatment lifelong prophylaxis is still a controversial   thing

Author Response

11/02/2023

Dr.L.S.M.Sigera

Manchester Fungal Infection Group,

Core Technology Facility,

The University of Manchester,

Manchester Academic Health Science Centre,

Manchester, UK.

Dear Reviewer of the Journal of Fungi,

Invasive aspergillosis after renal transplantation (Manuscript ID jof-2189704)

I am submitting a revised manuscript for consideration for publication in the special Issue "Biology, Immunology, Epidemiology, and Therapy of Fungal Infections: A Themed Issue Dedicated to Professor David A. Stevens". The manuscript is entitled “Invasive aspergillosis after renal transplantation”.

We really appreciate your comments because they invariably improve the quality of the manuscript. All the changes were highlighted (In Blue)  and comments were answered in the original manuscript as track changes.

Its’ well written, nicely complied the published data.-Thank you for the comment

Study presents the main points:

  • Infection pulmonary aspergillosis  in renal transplant patient get infections after 6 months –Included
  • Can present without typical sign symptoms of lung –Included
  • It can present like space occupying lesion –Included
  • Infarctions, convulsion, tracheobronchitis endocarditis after dissemination and few others –Included
  • Like other IPA high GM index values predict high mortality –Included
  • Radiological finding: Most of the patient present well circumscribed nodules –Included
  • Pleural involvement is rarely seen –Included
  • Treatment: other than antifungals gamma interferon allows both the patients and graft survival –Included
  • After treatment lifelong prophylaxis is still a controversial   thing –Included

Thank You

Yours sincerely

………………………..

Dr.L.S.M.Sigera
